# All-optically controlled phased-array for ultrasonics

Rahul Goyal [1,2], Oscar Demeulenaere [1,2], Marc Fournelle[3], Athanasios G. Athanassiadis [1,2] & Peer Fischer [1,2,4,5] ✉

The ability to dynamically shape ultrasound fields is critical for emerging applications in therapeutic ultrasound, particle manipulation and tissue engineering. While existing phased arrays provide beam steering for imaging, these newer applications require higher intensities. This complicates the electrical driving and ultimately limits the array size and spatial complexity of the field. Here, we introduce a scalable architecture for driving phased arrays using a single power source and light-responsive analog phase shifters. Compared to conventional arrays, which drive each channel independently, our device only needs one amplifier. Moreover, the phase shift can be continuously varied between $\pm\pi$ based on light intensity. Using our phase shifter, we demonstrate dynamic, multi-focal ultrasound beams, fast beam steering, and spatially-complex beams including acoustic vortices. Because of its simple, analog design, optical addressing, and superior phase control, this architecture paves the way for very large transducer arrays and the generation of high-intensity, spatially-complex ultrasound fields.

Ultrasonic phased array transducers (PATs) consist of many individual sound-generating elements, each with dimensions and spacing typically on the order of an acoustic wavelength. Most PAT geometries allow wide beam steering and focusing in real-time for critical applications in industrial testing and medicine, such as diagnostic imaging and ultrasound therapy. However, the acoustic pressure fields that can be synthesized by PATs are limited by their relatively low element count, severely constraining the spatial complexity of pressure fields that can be generated with them. Recently, it has been shown that acoustic holograms[1] can generate far more complex fields than PATs because of their higher effective number of pixels (elements) and thus higher information density[2]. Holograms, for instance, have been developed to produce complex fields[3] that can be used for manipulation of objects, such as cells in 2D[4,5] and 3D[6].

While PATs are sufficient for imaging, applications for ultrasound therapy are more challenging and often require the generation of high intensity, spatially complex fields to target small and complex-shaped regions, for instance in the brain[7–11]. However, extending PATs to large element counts, which is needed for complex field synthesis, is highly nontrivial, as highlighted by the fact that arrays do not exist larger than at most a few thousand elements. One of the main challenges in scaling traditional PAT architectures lies in the electronic driving of the individual channels, where each channel requires its own high-voltage pulse circuitry.

Despite over 50 years of development that have led to more reliable and efficient PAT hardware[12,13], this multi-channel sequential architecture for individually addressing the transducer elements has remained fundamentally the same. Recent technical developments, such as micro-beamforming[14] row-column addressing[15], or sparse arrays[16], have significantly improved the ability to address more pixels in phased arrays. However, these techniques inherently remove degrees of freedom to transmit field patterns and thus cannot easily be translated to project complex fields for holography. And while new, low-voltage transducers designed for imaging[17] could relax some of

[1]Max Planck Institute for Medical Research, Heidelberg, Germany. [2]Institute for Molecular Systems Engineering and Advanced Materials, Heidelberg University, Heidelberg, Germany. [3]Ultrasound Department, Fraunhofer Institute for Biomedical Engineering, St. Ingbert, Germany. [4]Center for Nanomedicine, Institute for Basic Science (IBS), Seoul, Republic of Korea. [5]Department of Nano Biomedical Engineering (NanoBME), Advanced Science Institute, Yonsei University, Seoul, Republic of Korea. ✉e-mail: peer.fischer@mr.mpg.de

the requirements for imaging array electronics, therapeutic applications require high powers that are currently only possible using electrical architectures that scale poorly with size[18]. Although solutions have been developed for high-power electrical driving of therapeutic arrays[19], such solutions are specific to point-focused geometries and would be insufficient for significantly more complex field shapes for diverse applications. There is therefore a strong need for a new architecture capable of independently driving thousands of transducer elements in parallel with high-power, rapid update rates, and tight timing or phase control to allow for precise field synthesis.

Here, we introduce a light-controlled analog phased array architecture—the optically programmable array of transducers (OPAT)—that can deliver high-power electrical waveforms with precise phase control, and which can be programmed optically to generate dynamic ultrasonic wavefronts. The key electrical innovation in our architecture is our use of a balanced dual-cascaded RC network whose photosensitive resistors provide optical control over the electrical output phase. In contrast to wired electrical approaches, optical addressing[20] can provide parallel, scalable[21], and wireless control. Thus, this approach is attractive for applications including non-destructive testing[22], therapeutic ultrasound, and neuromodulation[23–25], which would all benefit from more accurate field-shaping with high-power ultrasound.

Our OPAT architecture, as shown in Fig. 1, requires only a single amplified electrical input signal that is distributed and independently modulated for each transducer element, eliminating the need for independent pulse circuitry and amplifiers for each channel. Moreover, because the signal driving each channel is inherently phase synchronized, there is no need for a separate clock, and all the phase modulation can be accomplished via the optical inputs. Finally, our electronic architecture has a wide operational bandwidth, supporting the use of transducers over a wide range of center frequencies.

The core element in the OPAT architecture is the light-activated phase shifter (LAPS), which is an analog circuit for high frequency operation based on photoresistive elements coupled with passive electronic devices in a cascaded architecture. The electrical circuit converts an optical intensity into a precisely phase shifted electrical signal. The real-time phase is controllable by varying the intensity of the illuminating light, which can be done for all the channels in parallel by using standard, commercially available projectors. The concept of controlling the phase of each transducer in the array separately with light is illustrated in Fig. 1: light patterns are dynamically projected (Fig. 1A, B) onto the photoactive layer (Fig. 1C), which consists of photoresistors (Fig. 1D) that provide an intensity-dependent phase shift (Fig. 1E), via the resonant LAPS circuit (Fig. 1F). The conductivity change of the photoresistor allows for a continuously tunable phase shift of up to $2\pi$ (Fig. 1E) independently for each transducer element. The LAPS circuit (Fig. 1F) can be used to shift the phase of the driving signal across a wide ultrasound bandwidth ranging from under 100 kHz to over 10 MHz (shown in Fig. S2 of the Supplementary Information). The phase shifted signals are then sent to the transducer array (Fig. 1G, H) and converted into ultrasound. These dynamically-modulated wavefronts (Fig. 1I) then propagate through the medium to form complex, dynamic ultrasonic fields (Fig. 1J) that can be used for manipulation or therapy.

Below, we introduce the OPATs and LAPS architecture, describe their performance theoretically, and validate their light-coupled phase shifting behavior experimentally using both electrical measurements and acoustic measurements. We demonstrate light-driven acoustic field synthesis using our OPATs to drive a $11 \times 11$ transducer array, optically defining single- and multi-focus output fields as well as vortex beams. Because the system is driven by a light projector, we can dynamically project a movie of phase patterns to generate fast, reconfigurable ultrasound wavefronts. We employ a digital light projector to generate the optical patterns, but other optical projection

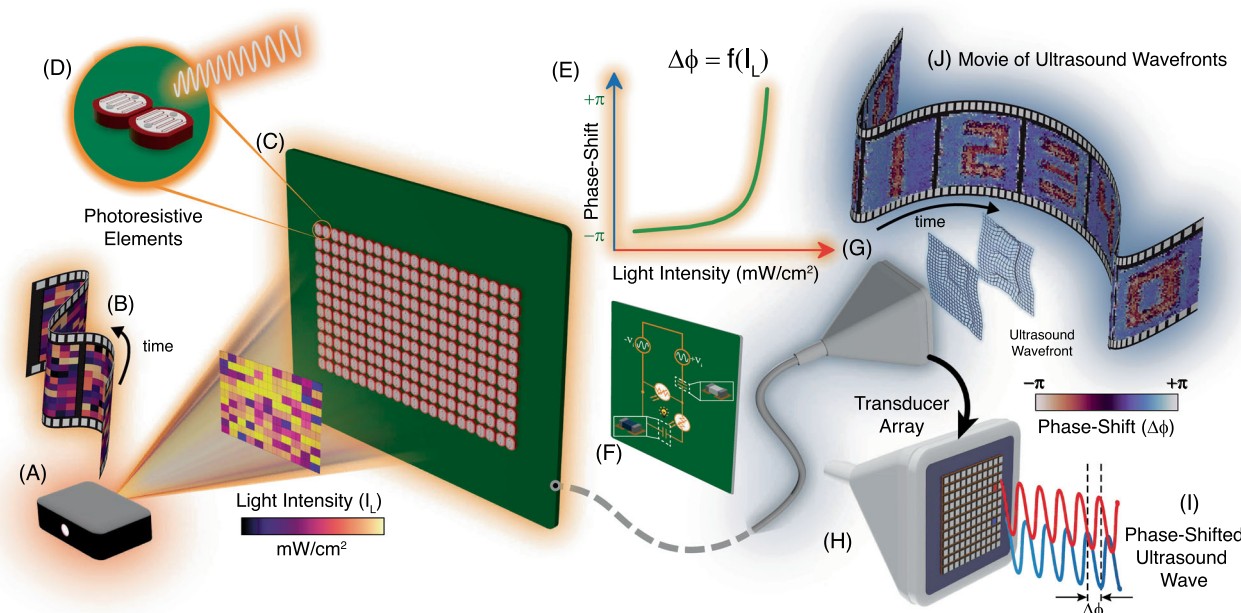

**Fig. 1 | Optically programmable array of transducers (OPAT) uses light to shape ultrasonic fields.** The OPAT is programmed by light intensity patterns ($I_L$), which are illuminated from **A** the projector unit that plays a **B** time-varying image pattern. The pixel units in the OPAT are individually and independently addressed by **C** illuminating light and contains **D** an optically active network based on dual-cascaded photoresistors. The time domain characteristics of the network permits operation over a wide frequency range, and the **E** illumination intensities produce phase shifted ($\Delta\phi$) electrical signal that drives an individual transducer element (**F**). The optical control provides favorable scalability to **G** larger-sized transducer arrays, which can be simply plugged to the network to generate **H** dynamic phase distributions without the need for complex electronic circuits or arrays of power amplifiers. The **I** relative phase shift ($\Delta\phi$) of the ultrasound waves emitted from the individual transducers **H** can therefore be tuned remotely. Under different light patterns, the programmable OPAT can generate various transmission phase distributions, thus creating complex ultrasonic wavefronts which are dynamically switched analogous to a **J** movie of ultrasound wavefronts.

methods, such as a digital micromirror device or a light-emitting diode (LED) matrix, could also be used to generate the illumination patterns. Our experimental results agree well with a numerical analysis over a broad operational bandwidth, highlighting the ability of our hardware to drive a wide range of typical ultrasonic arrays. Finally, we demonstrate the ability of our hardware to drive an 11 × 11 array with a single radio-frequency (RF) power input and a spatially resolved phase modulation up to $2\pi$. We show our architecture provides continuous control over element phasing in arrays, which translates into more precise and finer wavefront shaping while also allowing for the transmission of longer pulses with higher power, compared with conventional PATs. Moreover, the architecture is scalable such that we envision very large ultrasonic phased arrays, which are simply not feasible with conventional schemes for applications that also require high powers. Our results ultimately suggest that hybrid optoelectroacoustic devices can make for compact, scalable, and wirelessly tunable ultrasound instruments that provide unique complementary capabilities to existing ultrasonic phased arrays.

## Results

The architecture of the light-activated pixel element in the OPAT is presented in Fig. 2A. We have used the circuit to drive transducers at 0.7 MHz and at 2.25 MHz. For the latter, we used lead zirconate titanate (PZT) piezoelectric disks as the ultrasound transducers, which are 3 mm in diameter, 1 mm thick, and have contact electrodes on the top and bottom surfaces. The transducers are driven at the frequency of 2.25 MHz with a peak-to-peak voltage $V_{PP}$ of 82.5 V and a maximum current of 12.7 mA. The transducer under the operation of a high frequency driving voltage is modeled as a capacitor, where the capacitance $C_T$ is in the range from 75.3 to 80.1 pF. We constructed a balanced

dual-cascaded RC network, which can be interpreted as a second-order system attributed to the two reactive components in the structure. With this configuration, and without manipulating the piezoelectric element itself, the effective time constant of the driver circuit can be changed by tuning the resistance of the embedded resistors, resulting in the desired phase tunability of the ultrasound wave. Crucially, the circuit can be used to obtain phase shifts from $-\pi$ to $+\pi$ at a fixed frequency $F$. This is in contrast to other electrical architectures, such as lumped component-based switched topologies[26] or impulse delay based phase topologies[27], which do not achieve a net phase shift of $2\pi$ and further operate with low tuning resolution (>22.5°). The phase shift $\Delta\phi$ of the LAPS with respect to the input electrical signal $V_i$ applied at the angular frequency $\omega = 2\pi F$ can be written as:

$$\Delta\phi = \tan^{-1}\left[\frac{\omega R(C_S - 2C_P)}{\omega^2 R^2 C_S C_P - 1}\right] - \tan^{-1}\left[\frac{\omega R(C_S + 2C_P + 2C_T)}{-\omega^2 R^2 C_S (C_P + C_T) + 1}\right], \quad (1)$$

where $R$ is the light-tunable resistance, $C_S$ and $C_P$ are the capacitance of LAPS capacitors, and $C_T$ is the capacitance of transducer element, as shown in Fig. 2A. In general, the impedance of a piezoelectric transducer is complex[14]. However, based on measurements on resonance (Supplementary Note 2 and Figs. S4 and S5), we approximate the magnitude of complex impedance of each transducer element as a purely capacitive reactance ($|Z_T| \approx |X_T|$). The mathematical formulation of $\Delta\phi$ as a function of complex impedance of transducer element is described in Note 1 of Supplementary Information.

The circuit diagram of the multistage LAPS architecture is shown in Fig. 2A. The combination of differential stages with a dual out-of-phase input voltage constructed a $\pi$ topology, which consists of two fixed capacitors ($C_S$ and $C_P$) and two variable resistors ($R$). In our

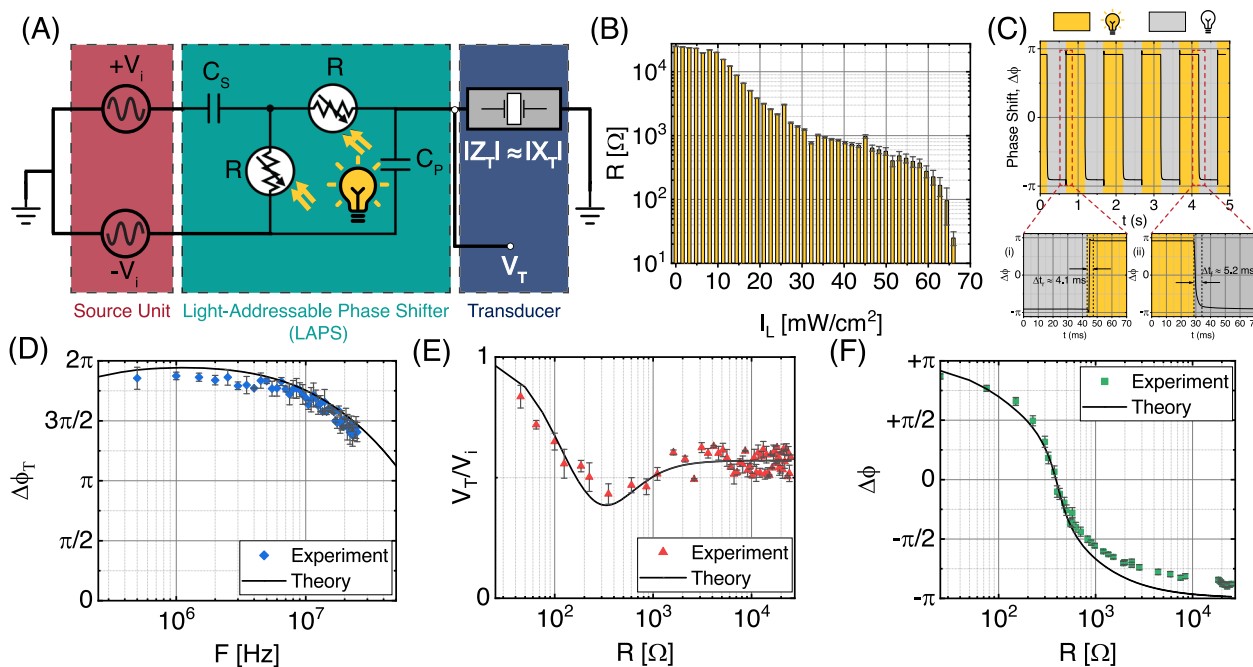

**Fig. 2 | Electrical architecture with the measured and calculated performance of the light-activated phase shifter. A** Simplified circuit diagram of the programmable pixel unit where the loaded transducer element has a complex impedance of $Z_T$, which we approximate by the capacitive reactance $X_T$. **B** Electrical characteristics of the photoresistive element embedded in the pixel unit, where the resistance R is measured for different values of light intensity $I_L$. **C** The phase shift of the emitted ultrasound wave when the light is switched on ($I_L = 65.7$ mW cm$^{-2}$) and off at 1 Hz. Insets (i) and (ii) show zoomed views of the rise and fall times $\Delta t_r$ and $\Delta t_f$, respectively. **D** Experimentally measured range of the total phase shift $\Delta\phi_T$ in the electrical signal presented for several different frequency values $F$. The

experimental results corresponds to the operation of transducers at 2.25 MHz where data are presented as mean value ± standard deviation in the mean value of the phase shift. **E** The experimental and theoretical curves of the relative amplitude of electrical signal presented for several different values of $R$. The resistance is controlled by the light intensity $I_L$. The data are presented as mean value ± standard deviation in the mean value of the relative amplitude. **F** The experimental and theoretical curves of the phase shift obtained in the ultrasound wave presented for several different values of $R$. The data are presented as mean value ± standard deviation in the mean value of the phase shift.

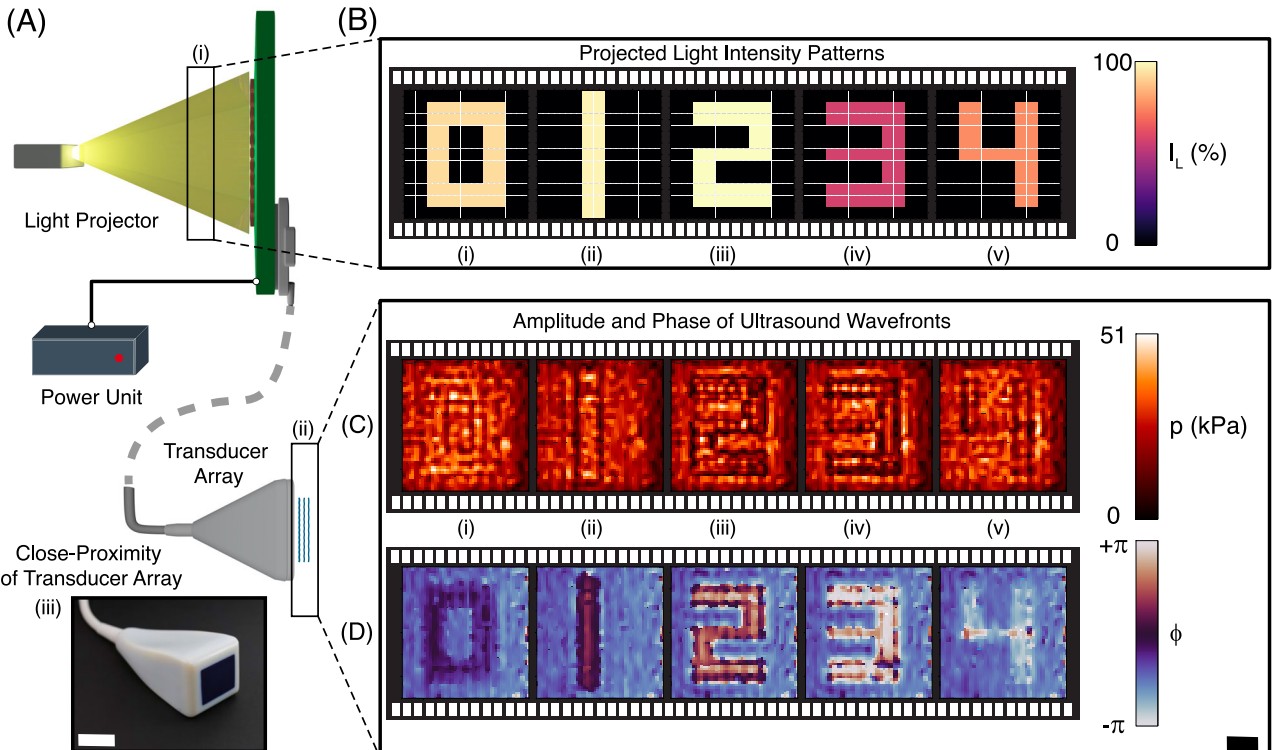

**Fig. 3 | Phase mapping of ultrasound wavefront based on the projected light intensity patterns. A** Schematic of setup: light projection unit for illuminating defined intensity pattern (i), single power source unit, and transducer array for emitting ultrasound wavefront (ii). (iii) The photograph of the integrated transducer operated at the frequency of 0.7 MHz (scale bar 30 mm). The hydrophone scan is performed 1.5 mm above the transducer array. **B** The projected light intensity patterns represented by **A** (i) which are spatially structured to describe the formation of numerical digits (i) "0", (ii) "1", (iii) "2", (iv) "3", (v) "4". The maximum projected intensity is 65.7 mW/cm². **C, D** (i)–(v) The spatial distribution of the amplitude and the achieved phase shift in the pressure wave recorded by the hydrophone. The scan area is 31 × 31 mm. Scale bar indicates 7.5 mm.

implementation of the LAPS we used a photoactive element (NSL-19M51, Advanced Photonix) with low capacitive charge storage and a high resistance ratio. These characteristics are necessary to achieve high phase tunability in a wide frequency band of ultrasound operation. The photoresistor's resistance $R$ can be varied continuously from 24.7 k$\Omega$ in the dark state to 54.5 $\Omega$ when it is illuminated with a light intensity of 65.7 mW/cm² (Fig. 2B). The change in resistance causes a shift in the circuit time constant leading to a change in phase for a fixed-frequency driving signal. A continuous $2\pi$ phase shift is achieved by cascading two stages of series RC network such that each individual stage can achieve a phase shift from 0 to $\pi$ at an operating frequency $F$. The electrical phase shift is accompanied by a change in voltage as shown in Fig. 2E, F. Experimental measurements of the transmitted voltage amplitude and phase from the LAPS are in excellent agreement with theoretical predictions across the range of photoresistor resistances. The shifts in voltage amplitude and phase result in a phase shift in the ultrasonic wavefront, as demonstrated by our hydrophone measurements of the pressure produced by a single piezo element (Fig. 2).

In order to provide real-time spatial control of ultrasonic beams, the circuit must be able to rapidly change the applied phase shift. The refresh rate of the phase reconfiguration is determined by the switching speed of the cascaded network. To experimentally determine the response times, we modulated the incident light and determined the rise time $\Delta t_r$ and fall time $\Delta t_f$ needed for the electrical phase to shift from $-\pi$ to $+\pi$ to $+\pi$ to $-\pi$, respectively, as shown in Fig. 2C. The measured response times are $\Delta t_r = 4.1$ ms and $\Delta t_f = 5.2$ ms. These switching times suggest that OPAT permit signal modulation at a rate of 100 Hz, which we demonstrated experimentally as shown in Fig. S12 of the Supplementary Information.

The performance of the phase shifter is independent of capacitive loading, making it effective for driving different transducers in different frequency ranges. In order to model the effect of different transducers, we numerically model the circuit response with different ranges of load capacitance up to 500 pF and found that the total phase shift remains constant (Fig. S3 of the Supplementary Information). Then, to verify the ability of the circuit to apply a $2\pi$ phase shift for different transducers across a wide range of center frequencies, we measure the phase shifting bandwidth by substituting the transducer element by an equivalent non-polarized thin film capacitor. The LAPS circuit provides a total phase shift of $2\pi$ in a wide frequency band ranging from 0.1 to 10 MHz, as demonstrated in Fig. 2D. Thus, the device facilitates controlled realization of sophisticated wavefronts by simple light illumination, with the benefits of analog control of the phase shift and complete phase modulation from $-\pi$ to $+\pi$.

To demonstrate the operational principle of our system using light to control a complex ultrasonic wavefront, we use the OPAT to drive an 11 × 11 transducer array. The transducer array operates at 0.7 MHz with an element pitch of 2.81 mm and a total acoustic aperture of 30.9 × 30.9 mm. The detailed information of our transducer array and electronics board have been provided in the "Methods" section and Supplementary Note 3, respectively. We projected several different light intensity patterns shown in Fig. 3A-(i) onto the LAPS. The associated phase shifted driving signals produced the spatially varying pressure fields, which we scanned with a hydrophone placed 1.5 mm from the transducer face (See Supplementary Note 5 and Fig. S11 for more details). The resulting pressure amplitudes and phases defining the emitted wavefronts are shown in Fig. 3B and C, respectively. The projected pressure fields consist of constant ultrasound amplitudes with small deviations observable near sharp phase boundaries as

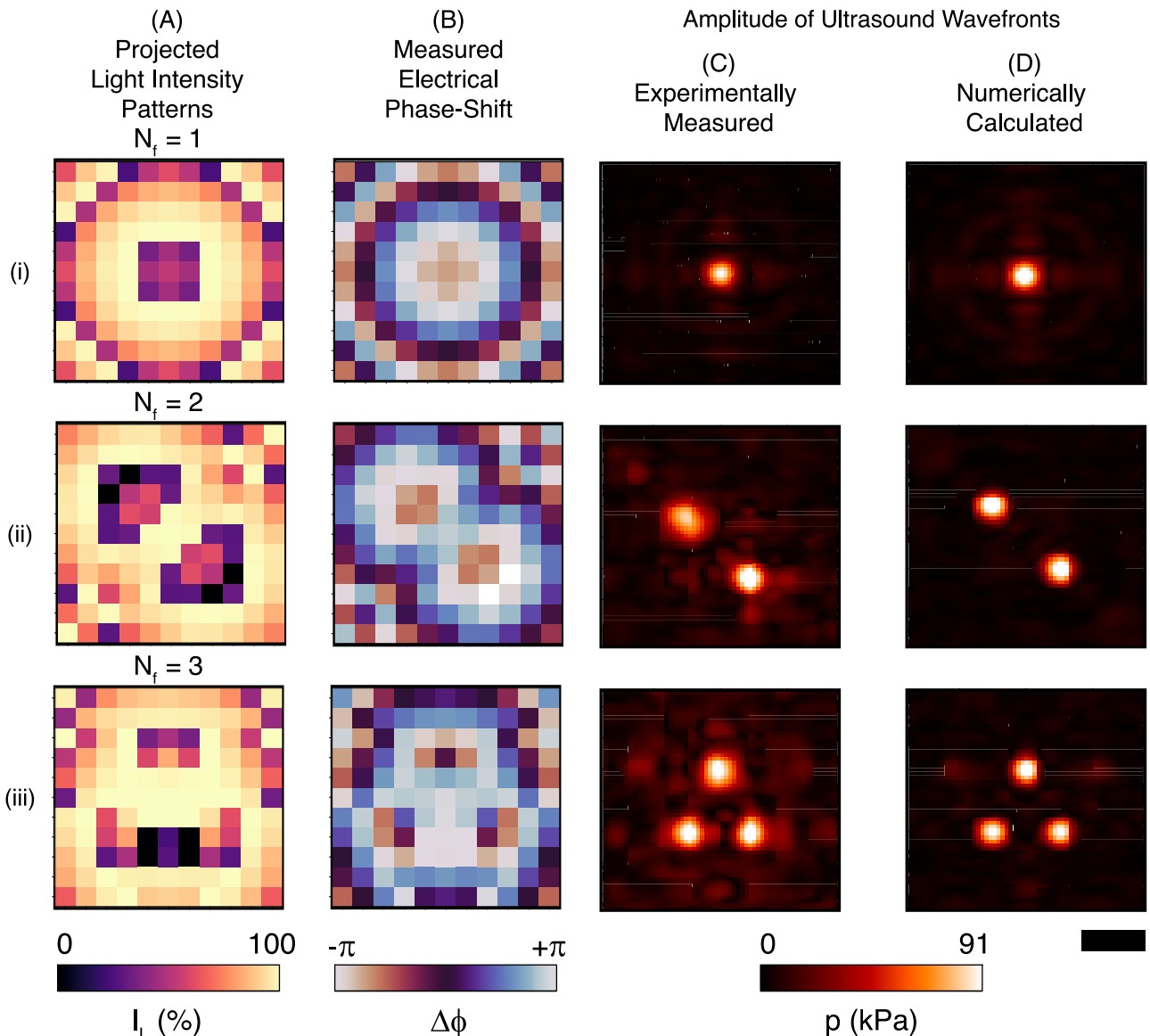

**Fig. 4 | Multi-focus generation and control with light. A** The spatially modulated light intensity pattern projected on the photoresistive elements to generate one (i), two (ii) and three (iii) acoustic foci. **B** The electrical phase shift ($\Delta\phi$) measured with respect to the input signal ($V_i$) driving the transducer elements and controlled by the projected light intensity ($I_L$). **C** The experimentally measured ultrasound pressure amplitude determined 50 mm from the surface of transducer array. **D** The corresponding numerically simulated amplitude based on the mapped electrical phase shifts. The scale bar is 5 mm.

expected. The phase on the other hand, is progressively shifted with each new digit by $\pi/3$ in accordance with the increased light intensity incident on the OPAT. Minor spatial variations within uniform phase regions can be observed in Fig. 3B, C, which we attribute to interference and diffraction between transducer output and the hydrophone (see Supplementary Note 6 and Fig. S13 for more details). Overall, the performance of a single LAPS scales well to an entire array, providing well-defined, controllable phase shifts across the entire acoustic wavefront.

Relatively low light intensities are needed for phase shifting, as even ambient light can activate photoresistors. The light intensities that are employed in the current system are smaller compared to the intensities of direct sunlight[28]. Nevertheless, we expect that a device based on our technique will contain an enclosed illumination setup. While we use a commercial projector, future developments could make use of micro-LED and microlens arrays, which would reduce the required light intensities and also permit for more compact designs.

To demonstrate the potential of the OPAT architecture for applications needing dynamic control of ultrasonic wavefronts, we sequentially project different patterns of light to switch between ultrasonic focusing configurations with $N_f = 1$, 2, or 3 focal points in a plane 50 mm from the transducer face, as shown in Fig. 4.

Three steps are necessary to compute the light intensity pattern required to generate a desired ultrasound pressure wavefront at a given distance from the transducer. First, from the desired ultrasound pressure wavefront, the necessary phase shift for each transducer element is computed. This step can be performed analytically for simple fields, or iteratively using an optimization algorithm for more complex fields (further details in "Methods" section). Then, from the phase value of the transducer element the resistance values of the photoresistive element are calculated, considering the electrical characteristics of the circuit, while accounting for the amplitude constraint of the transducer's terminal voltage. The target light intensity patterns, as shown in Fig. 4A, are then calculated based on optical characteristics of the photoresistive element (Fig 2B). The

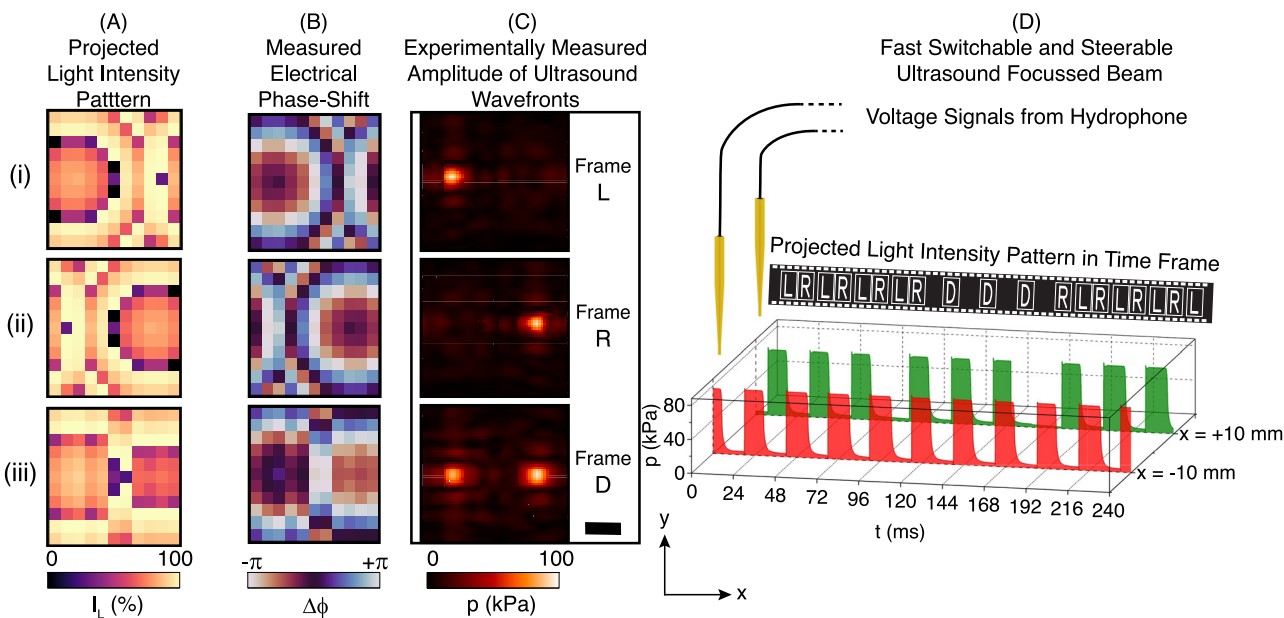

**Fig. 5 | Dynamic ultrasound wavefront shaping.** A fast-switchable multi-focus ultrasound wavefront is generated with the OPAT system. **A** (i), (ii), (iii) The spatial modulated light intensity distributions for constructing focused ultrasound at the points $x = -10$ mm, $x = +10$ mm, and $x = \pm 10$ mm, respectively. **B** The phase shift of mapped electrical signal actuating individual transducer elements independently corresponding to the light patterns of (i),(ii), and (iii). **C** (i),(ii), and (iii) The amplitude of ultrasound wavefronts recorded at the focal plane for light intensity pattern denoted by the frames L, R, and D, respectively, located at the distance of approximately 50 mm from the transducer surface. **D** The scale bar is 5 mm.

experimentally measured ultrasound pressure distributions are in good agreement with the numerical results as shown in Fig. 4C and D, respectively.

The proof-of-concept in Fig. 4 demonstrates a significant advancement in ultrasound technology, showcasing remarkably simpler way to shape ultrasound wavefronts. The power consumption per transducer element is notably low, requiring a maximum of 30 mW per channel, which is effectively equal to the acoustic power transmitted by each transducer pixel. Of this power, 77% is used for the optical illumination, which we did not optimize (see Supplementary Note 4 for details). Therefore, it is likely that with further design adaptations, such as integrated optical illumination could significantly reduce the parasitic power required by our phase shifting hardware.

Light-addressed phase control represents a crucial step towards scalable and compact ultrasonic systems. The detailed description of optical and electrical power consumption is provided in Supplementary Note 4 and Fig. S9 of Supplementary Information. It was predicted from the equivalent circuit model that the required driving voltage would need to be approximately doubled to achieve the same pressure magnitude as when connected to the LAPS circuit. This highlights an area for future optimization. The electrical architecture could be refined to bridge this gap, potentially through improved impedance matching, more efficient power delivery systems, or novel circuit designs. Such optimizations could further reduce power consumption or increase output pressure, enhancing the overall performance and efficiency of the ultrasonic system.

To demonstrate the capability of using an optically controlled phased array for rapid wavefront shaping, we switch between projecting ultrasound beams with a focus on the left (L, $x = -10$ mm), the right (R, $x = +10$ mm) or a double focus (D, $x = \pm 10$ mm) (Fig. 5). The fast-switchable operation of the OPAT is recorded simultaneously by two hydrophones positioned at each of the focal points ($x = -10$ mm and $x = +10$ mm) as shown in Fig. 5D. A schematic of the setup is shown in Fig. S11C of the Supplementary Information. The left focus, right focus and the double focus are switched, as schematically shown in the movie strip at the top of Fig. 5D. Each frame corresponds to a light illumination of the corresponding pattern (Fig. 5A) for 17.5 ms. The corresponding ultrasound pressure from the hydrophone is displayed in Fig. 5D at the bottom. As the projected light pattern is modulated, we clearly observe synchronized, rapid, and repeatable focus switching. Such phase shifting performance extends over a wide frequency range as well, allowing for the transmission and phase shifting of relatively short pulses. By transmitting single-cycle pulses through the array, we observe that the circuit applies a nearly constant phase shift across the operating band 0.2–1.1 MHz. Therefore, the OPAT can be used to transmit and apply a constant phase shift to broadband pulses with energy concentrated in this band (see Supplementary Note 7 and Fig. S14 for further details).

Along with dynamic wavefront modulation, our system provides the capability to project spatially complex wavefronts by phase engineering. An acoustic vortex beam is a prominent example, where the helical phase distribution is especially useful for particle manipulation[5,29,30]. Methods that have been used to generate an acoustic vortex beam include static phase masks[31] and structured transducers[32]. Here, we show that OPATs can produce helical phase distributions and generate vortex beams, as shown in Fig. 6A. The projected light intensity pattern was computed to generate the target phase gradient as shown in Fig. 6B-i, and the mapped distribution of electrical phase shift is shown in (Fig. 6B-ii). The pressure fields were measured with a hydrophone (Fig. 6B-iii) and compared to numerical predictions of the far-field acoustic vortex (Fig. 6B-iv). The amplitude distribution and the phase gradient are both in good agreement with the numerical simulation, demonstrating the ability of the OPAT architecture to produce spatially complex ultrasonic wavefronts.

## Discussion

Next-generation biomedical applications of structured ultrasound fields require the delivery of high acoustic power into spatially complex shapes. In order to meet these needs, it will be necessary to use transducers with both larger apertures and higher element counts than are currently used for imaging, which remain a challenge to fabricate and have until now been impractical to drive electrically. Here, we have

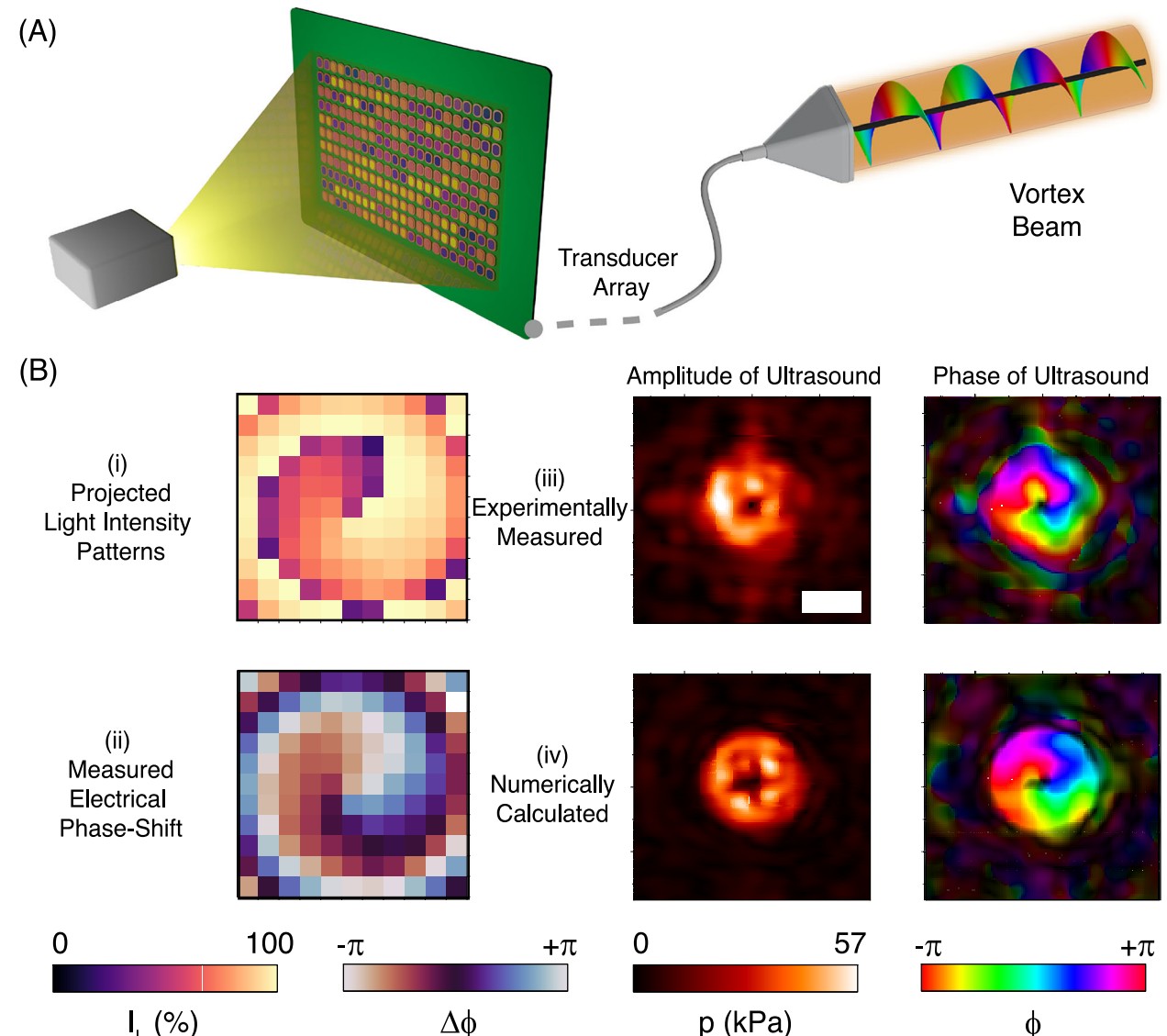

**Fig. 6 | Generation of ultrasound phase gradient in the vortex beam wavefront.**
**A** Schematic representation of vortex beam generation by projecting the corre-
sponding spatial light intensity distribution. **B** (i) The spatially modulated light
intensity, which resulted in the (ii) electrical phase shift mapped for the transducer
array. The experimental (iii) and numerically (iv) obtained amplitude and phase
maps of the acoustic pressure pattern, which is measured in a plane located at a
distance of 60 mm from the transducer surface. The measured maps are consistent
with the theoretical calculations. The scale bar is 7.5 mm.

introduced a new architecture for electrically driving phased arrays
using programmable light intensity patterns. The key innovation in our
design is a balanced dual-cascaded RC network whose photosensitive
resistors enable wireless and massively parallelizable control over the
electrical output phase. This design allows us for the first time to
optically modulate the driving phase for each array element con-
tinuously between $-\pi$ and $\pi$. Moreover, our architecture can inde-
pendently drive all transducer elements in an array using only one
amplified RF signal, dramatically simplifying the operation of large
transducer arrays. The present system is fast (100 Hz) and scalable, as
the light-triggered phase switching can be realized with low-intensity
optical projection (66 mW/cm²). Moreover, the circuit is able to
accommodate different transducer capacitance and a broad frequency
range from 100 kHz to 10 MHz. We use our architecture to drive a
standard $11 \times 11$ imaging array and generate complex wavefronts,
demonstrate switchable focusing, and project vortex beams.

Because of its parallel addressing, single power source, and load-
independent performance, the optical phase shifter introduced here
should scale favorably to permit the operation of very large element

transducer arrays and thus dramatically improve the ability to form
dynamic complex ultrasound fields. To effectively drive larger arrays with
a compact and flexible footprint, the optical modulator could be inte-
grated, for instance with the help of micro LED arrays in conjunction with
a micro-lens array for efficient light collection. The modulator circuit
could then be directly integrated with an emitter transducer via flip-chip
bonding[33,34]. Such an integrated architecture could then support further
scaling without requiring more complex interconnects and electronics,
paving the way for new applications in biomedicine, including
ultrasound-assisted tissue engineering and ultrasound therapy.

## Methods
### Electrical driving of transducers with OPAT
To drive ultrasonic transducers, the OPAT is supplied with two sets of
synchronized RF voltage signals. The first is denoted by $+V_i$ and the
other is a $\pi$ phase shifted signal represented as $-V_i$. The frequency of
the driving signal was either 2.25 or 0.7 MHz, depending on which
transducer was being used for the experimental demonstration. In our
experiments, the positive and negative source signals were generated

separately by a single function generator (AFG 1062, Tektronix), and each amplified by their own low-impedance 50× voltage amplifier (WMA-300, Falco Systems). It would be possible, however, to split a single amplified RF signal and use a phase shifter on one branch of the signal to provide both $\pm V_i$ to the OPAT.

### Measurement of ultrasound wavefronts

Ultrasonic pressure fields were mapped using a 0.5 mm-diameter needle hydrophone (HNR0500, Onda Corporation) scanning in a water tank. An open 3D printed water tank (175 mm × 175 mm × 75 mm) made from a UV-cured polymer (VeroClear, Stratasys) was filled with deionized water and the transducer array was placed in acoustic contact with the water. For experiments with the 11 × 11 array, the array was coupled to the outer surface of the water tank using a thin layer of vacuum grease (Dow Corning), projecting through the wall and into the tank. For experiments with the PZT disks, the piezoceramics were mounted onto a printed circuit board, which was waterproofed with a spray coating of dielectric strength 48 kV/mm (RS Components GmbH, Germany) and the water tank was coupled to the piezos using vacuum grease.

Pressure waves from the transducer produce a voltage in the hydrophone, which was digitized using a USB oscilloscope (Picoscope 5000 Series, Pico Technology) and saved on the computer. The time-domain voltage data were converted to a pressure using the manufacturer-provided calibration. For accurate measurements, a 25-cycle sinusoidal pulse was used to drive the OPAT. The hydrophone correspondingly recorded a pulse, which was converted to amplitude and phase data at the center driving frequency using a Fast Fourier Transform.

To measure the spatial variation in the pressure field, the hydrophone was mounted on a computer-controlled three-axis motorized translation stage (MTS25-Z8, Thorlabs GmbH). The scan area in the imaging plane was 20 × 20 mm, with a lateral resolution of 0.37 mm. 25 pulse measurements were averaged at each point of the imaging plane to increase the signal to noise ratio. A typical scan took 3–4 h.

### Computation of source phases for complex wavefronts

We used the Iterative Angular Spectrum Approach (IASA) to compute the relative phase shift for individual transducer elements required in generating the programmed wavefront[1,35,36]. The phase distribution of the transducer elements is determined by iteratively propagating a wave from the target plane to the transducer plane and back, with amplitude constraints being applied in each plane. In our adaptation of the IASA algorithm, we applied an amplitude constraint per pixel to match the experimentally measured phase shift-dependent amplitude. The amplitude constraints for every iteration are set separately and explicitly. The algorithm converges in a few tens of iterations to yield the desired phase map. We performed 50 iterations for every wavefront to accurately compare the performance of OPATs for different applications.

To set the transducer phases for fields with higher resolutions than the driving array, we subdivided the transducer elements into multiple pixels and assumed a uniform phase output of all pixels corresponding to a single element. The relative phase shift for individual transducer elements is then calculated by averaging the obtained phase shift for all pixels comprising one element. The algorithm provides good experimental results, which are consistent with the theoretically predicted wavefronts.

The IASA-computed phase distribution is then converted to a light intensity using a look-up table based on the experimentally measured curves provided in Fig. 2.

### Bandwidth measurement for the light-addressable phase shifter circuit

It is known that for an RC circuit the phase shift angle of the output electrical signal changes as a function of operational frequency, and the rate of change depends on the temporal characteristics of the circuit[37]. Hence, to manipulate the phase of ultrasound waves over a wide frequency range, one of the most effective approaches is to design a band-pass circuit with an appropriate Q-factor. Since piezoelectric transducers can be modeled as a frequency-dependent capacitive load $C_T(F)$[38], we characterized the OPAT frequency-dependent performance by substituting the transducer by equivalent non-polarized thin film capacitors. We measured the phase modulation depth ($\Delta\phi_T$) of the output electrical signal ($V_T$) with respect to the input electrical signal ($V_i$) for different capacitors corresponding to specific driving frequencies. For each capacitance $C_T$, the modulation depth was determined by varying the intensity of incident light across its full range.

### Transducer array

The matrix array transducer was custom made in an 11 × 11 element configuration. First, a 1–3 PZT composite material was generated from bulk PZT using a dice and fill technique. The composite plate was then applied a backing carrier material and the matrix structure was generated with a new dicing step. Dicing was performed with a kerf of 250 μm and a resulting pitch of 2.81 mm. Individual contacting of the elements was generated from bottom side. The array gaps were filled with an epoxy material and a common ground electrode was applied using a sputtering process on the top side. A matching layer was then applied for optimization of transducer bandwidth. After housing integration, a protective coupling layer made of soft silicone was applied. The phase shifted signals were sent to a 11 × 11 transducer array via a multi-micro coaxial cable terminated with an ITT Cannon connector[39], which could be connected to the OPAT for transfer of electrical power. The capacitance $C_T$ of individual elements is measured to be in the range 301.7–309.2 pF.

### Data availability

The experimental data generated in this study have been deposited in the Zenodo database under accession code https://doi.org/10.5281/zenodo.16758255. Contact Peer Fischer for further requests of any other additional materials.

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

## Acknowledgements

This work was supported by the European Research Council under the ERC Advanced Grant Agreement HOLOMAN (No. 788296). The authors thank the Deutsche Forschungsgemeinschaft (DFG, German Research Foundation) under Germany's Excellence Strategy via the Excellence Cluster "3D Matter Made to Order", EXC-2082/1-390761711 (O.D. and P.F.).

## Author contributions

P.F. conceived the project. R.G. developed the opto-mechanic hardware and carried out the experimental work, including designing of electronics architecture, modeling, acoustics simulations, and data analysis. O.D. helped in the data collection. M.F. provided the transducer array and helped with the electrical characterization. A.G.A. designed the acoustic scanning experiments and assisted in the data analysis and numerical calculations. R.G. and P.F. wrote the first draft of the manuscript, and all authors reviewed and edited the manuscript and Supplementary Information.

## Funding

## Competing interests

The authors declare the following competing interests: R.G., A.G.A., and P.F. are listed as inventors on a patent application related to an optically controlled phase shifter for phased array ultrasonics. They have no additional financial or non-financial competing interests. M.F. and O.D. declare no competing interests.
