## [Transparent Peer Review file · Nature Communications]

All-Optically Controlled Phased-Array for Ultrasonics

Corresponding Author: Professor Peer Fischer

Version 0:

Reviewer comments:

Reviewer #1

(Remarks to the Author)

The paper presents an approach to modulate the phase of an excitation signal for individual ultrasonic emitters. Two photo-resistors are illuminated from a projector to individually control the phase-delay of a common excitation signal, the photo-resistors can be placed on an array layout and addressed in parallel by an image from a projector.

The concept is simple and well-developed. However, I am not sure about how useful it is. In simple words, it cannot be used to do things that were not achieved before by other means. Also, in the current implementation the phase-modulation is separated from the emitting array and the signals delivered by a cable.

The sentence of the paper "The main challenge in scaling traditional PAT architectures lies in the electronic driving of the individual channels." should be toned down. It is one of the main challenges, but not the main.

As I understand now, the phase-modulation layer is separated and the phased-delayed signals are delivered to the emitting array with a cable.

The approach has some advantages, it seems easy to replicate. But if the modulator is separated from the emitter array, then common electronics are not that complicated or hard to be used. Also, the projector introduces some disadvantages, for example, the required extra space and how fast the emissions can be modulated.

"signal modulation at a rate of 100Hz which we demonstrated" I do not consider this fast-switching. Switching speeds of traditional electronic drivers can be around kHz for specific applications.

More detailed pictures of the modulation layer/circuit should be present. Sizes, fabrication process. It is clear that can modulate at 100Hz but how much power is wasted when driving the emitters?

Is there any possibility of future fabrication of this light-controlled modulator in the emitters? That should be discussed in more detail.

Reviewer #2

(Remarks to the Author)

This manuscript describes a novel way of optically modulating an ultrasound matrix array. It has the advantage of supplying a single drive waveform which can be optically modulated by a digital projector. It is well written and explanations are clear in general. Some experimental details are missing.

lines 36-38: The authors may be unaware of matrix arrays using thousands of elements using micro-beamformers and advanced techniques. These methods are described on Philips Ultrasound website and in Chapters 7 and 10 of the book Diagnostic Ultrasound Imaging: Inside Out, (2014)

Line 79 reference [24] does not describe dispersion correction but dispersion measurement. An explanation of absorption and dispersion listing original references can be found in Chap. 4 of the previously mentioned book.

Lines 124-131, Equation 1, Fig. 2, line 323 [34]

The assumption that the transducer impedance can be represented by a capacitor is not accepted as a valid equivalent circuit model. Ref [34] is only valid for a piezoelectric in air and not acoustically loaded. In general, a piezoelectric transducer

radiates into acoustic loads such as those used in both experimental arrays and described in lines 279-282, 333-344. Experimentally validated equivalent circuit models such as the Mason or KLM models are explained in Acoustic Waves by Kino or chapter 5 of the previously mentioned book. These models support acoustic loading on both sides, dielectric and absorption losses and harmonics (multiples of the fundamental frequency). In general transducer impedances are complex functions of frequency and have real and imaginary parts. Equation 1, however, appears to assume the transducer impedance is a pure capacitor. Whether this is a reasonable approximation in general remains to be proven. Can you show that this approximation holds for the specific transducers in your experiments? Is this assumption reasonable for very small matrix elements? The circuit in Fig 2 and Equation 1 and supplement 1 can be adapted to be more realistic by adding a real component RT .

Lines 322-329: Based on the assumption that transducer impedances can be solely represented by capacitors, tests were run on thin film capacitors at different frequencies. These would not constitute a comprehensive study, given that transducers come in many sizes and frequencies and actually have complex impedances. Can you provide more specifics on your experiments like the range of capacitances and frequencies studied? Are these experiments supporting the claim that the circuit supports frequencies from 100kHz to 10 MHz? Furthermore, are these experiments the source of data shown in Fig. 2C? These connections are not clearly stated in the manuscript.

Lines 159-165: The rise and fall times of the proposed method indicate that a 100 Hz frame rate or modulation replenishment can be achieved. The examples in the manuscript are implemented as a quasi-continuous wave methodology. Lines 286-287 confirm this as a tone burst of 25 cycles is used as a drive pulse. It is not clearly explained how broadband width time signals as stated in lines 75-79 can be achieved with these rise and fall times. Please elaborate.

Reviewer #3

(Remarks to the Author)

This is a really important contribution to the field of acoustic – and I don't say this lightly – particular for HIFU and object manipulation applications. Ever since the development of arrays and their widespread adoption, this issue of scalability has been a major limitation. The issue is that every element needs a new channel of electronics. Hence, cost scales almost linearly with number of elements. The present paper succeeds in overcoming this limitation and their device uses a single ultrasonic reference input along with optically controllable delays. This means that shaped light (from a light projector) can be used to create user-defined phase patterns. The "pixels" are then light sensitive resistors, which through a simple circuit, produce a phase delay on a piezoelectric element. The response time is fast (c. 5ms), leading to a demonstration that is claimed to have a 100Hz update rate. The results are clearly presented and show that the system operates as intended. I strongly recommend this work is rapidly published. I have a few minor points that need further clarification.

1) From Fig 1c it seems that the device is a phase-shifting device. It has this attribute over a wide range of frequencies. This is exactly what is needed for continuous signals or very narrow-band pulses. However, to delay a broad-band pulse requires a phase ramp as the different frequencies need to be delayed by different amounts (of phase) to keep the pulse compact. Hence, whilst the device has broad-band phase-shifting performance, it is not suitable for broad-band pulses. This distinction should be made.

2) Related to the above, I am unsure if the device can (as is suggested on P3) be used for time reversal? I think the caveat is needed that the signals being time reversed are narrow-band.

3) P5 – it is said that $65.7\text{mW}/\text{cm}^2$ of light is needed. Can the authors clarify if this is a safe level and perhaps provide some reference that readers can quickly follow, e.g. is this equivalent to a small hand torch?

4) Related to the above, if it is too sensitive then it might be hard to use in an environment with natural lighting, can the authors comment?

5) In figure 3 (and possibly fig 6) is it possible to quantify the phases obtained? Clearly there are observable phase differences in 3d, but there is also considerable variation. Can this be quantified and its source commented on?

6) There are a few very nice schematics of the apparatus, but it would be beneficial to have a photograph so the reader can understand the reality of the set-up.

Version 1:

Reviewer comments:

Reviewer #1

(Remarks to the Author)

The authors have addressed all my questions. I am not fully convinced that the proposed system should go into Nature Comms. In my view, this approach only solves the driving electronics, the wiring and emitters remain with the same issues. I still think existing electronic system can also be scalable and present multiple advantages compared to this system. In any case, it can result of interest to a wide variety of researchers giving that it is easy to replicate and applicable to different fields.

Reviewer #2

(Remarks to the Author)

The authors have addressed reviewer comments well.

Reviewer #3

(Remarks to the Author)

The authors have responded comprehensively to my comments, which were minor issues. Hence I am satisfied with the response and look forward to seeing this important work published. As I mentioned in my original review the importance of this work comes from the breaking of the current link between separate excitation channels and the number of array elements. This link means that as the number of array elements increases, so the cost goes up almost linearly. There are 1000-element systems in existence, but they are prohibitively expensive, and this limits their range of uses. With the concept described in this paper, the number of elements can increase to these levels without the commensurate increase in cost. Hence this opens up the possibility of the use of arrays with many elements in a much wider range of applications. For example, there are many uses of arrays in biomedical applications that are not being explored as the cost of entry is too high. A lower cost array system (using the ideas of this paper) would mean that this technology could be taken up by a much greater range of biomedical researchers.

We thank the reviewers for the constructive comments and suggestions to improve the manuscript. We have provided additional information, as requested, and revised the manuscript based on the feedback. Below we provide a point-by-point response to all the comments.

Reviewer #1

1.1) The paper presents an approach to modulate the phase of an excitation signal for individual ultrasonic emitters. Two photo-resistors are illuminated from a projector to individually control the phase-delay of a common excitation signal, the photo-resistors can be placed on an array layout and addressed in parallel by an image from a projector. The concept is simple and well-developed. However, I am not sure about how useful it is. In simple words, it cannot be used to do things that were not achieved before by other means.

The reviewer asks whether our system can 'be used to do things that were not achieved before by other means.' While ultrasonic phased arrays are designed for use in both transmit/receive modes, our electrical architecture **is designed specifically to address gaps in the abilities of existing phased arrays to shape transmitted wavefronts**. Because of this more focused application, we are able to design a novel analog architecture that, in its current form, possesses several features that go beyond the current state of the art:

- We achieve a continuous phase modulation for arbitrary wave driving, where current systems only achieve limited discrete phase changes. The tuning resolution of current systems is low, as they operate with a phase resolution of > 22.5 degree ($\pi/8$), which is a significant impediment when it comes to wavefront shaping.
- Our setup can deliver longer pulses with higher power than can be transmitted with conventional PAT driving electronics.
- Our optically controllable architecture is scalable, which opens up opportunities that cannot be achieved with current systems.

Along with these advantages over phased array systems, our system improves on state-of-the-art dynamic holograms by providing continuous and arbitrary phase shifting over a broad range of frequencies, with an update rate 100x faster than previously demonstrated experimentally (doi:10.1002/adv.202104401).

We have stressed these points in the revised manuscript (line numbers are indicated and a highlighted version of the manuscript and the SI have also been provided for the review):

(L116) *Finally, we demonstrate the ability of our hardware to drive an 11x11 array with a single radio-frequency (RF) power input and a spatially resolved phase modulation up to 2π . We show our architecture provides continuous control over element phasing in arrays, which translates into more precise and finer wavefront-shaping while also allowing for the transmission of longer pulses with higher power, compared with conventional PATs. Moreover, the architecture is scalable such that we envision very large ultrasonic phased arrays, which are simply not feasible with conventional schemes for applications that also require high powers.*

1.2) Also, in the current implementation the phase-modulation is separated from the emitting array and the signals delivered by a cable. The sentence of the paper "The main challenge in scaling traditional PAT architectures lies in the electronic driving of the individual channels." Should be toned down. It is one of the main challenges, but not the main.

We have rephrased the statement to read:

(L49) *One of the main challenges in scaling traditional PAT architectures lies in the electronic driving of the individual channels.*

1.3) As I understand now, the phase-modulation layer is separated and the phased-delayed signals are delivered to the emitting array with a cable. The approach has some advantages, it seems easy to replicate. But if the modulator is separated from the emitter array, then common electronics are not that complicated or hard to be used. Also, the projector introduces some disadvantages, for example, the required extra space and how fast the emissions can be modulated. “signal modulation at a rate of 100Hz which we demonstrated” I do not consider this fast-switching. Switching speeds of traditional electronic drivers can be around kHz for specific applications.

The reviewer is right that existing drive electronics with cabled transducers effectively address the needs of many current applications, particularly in imaging. However, as we argue in the manuscript (L56), such architectures cannot be scaled up effectively to larger pixel counts, which would be necessary for higher-fidelity field-shaping (e.g. in therapeutic or particle manipulation applications). While more recent solutions (including sparse arrays and low-phase-resolution steering) have enabled the deployment of larger therapeutic systems, these are geared toward the production and manipulation of point foci, and do not translate well to the phase manipulations necessary for more complex field shapes.

While our proof-of-principle study makes use of a large projector and hand-assembled circuits, this is not a fundamental limitation, and it is quite reasonable to expect that the system could be made significantly more compact using integrated on-chip electronics and micro-LED arrays.

Regarding the switching speeds, we again draw a distinction between the aims of our system and those of traditional imaging systems. While existing imaging systems can indirectly project short pulsed excitation fields (plane wave, focused beam) with up to tens of kHz rates, such systems are not geared towards projecting complex field patterns, particularly not with higher powers or longer pulses. On the other hand, existing systems for dynamic holographic field shaping - which have been shown to project fields of significantly higher complexity and quality than can be projected with PATs (doi: 10.1103/PhysRevApplied.12.064055) – have to date been limited to refresh rates up to about 1 Hz (doi: doi:10.1002/adv.202104401). **In this context, our architecture improves upon the state of the art by 100x which is a significant scientific advancement in the field of dynamic shaping of ultrasound wavefronts.** Moreover, this speed limit is set by the RC time constant of our phase shifting circuit and is therefore independent of the array size for our architecture.

We have extended our discussion to address the aspects raised by the reviewer regarding the speed and electrical complexity in the following in the updated manuscript:

(L60) *... therapeutic applications require high powers that are currently only possible to drive using electrical architectures that scale poorly with array size. Although solutions have been developed for high power electrical driving of therapeutic arrays [DOI: 10.1088/0031-9155/61/17/R206], such solutions are specific to point-focused geometries and would be insufficient for significantly more complex field shapes for diverse applications.*

1.4) More detailed pictures of the modulation layer/circuit should be present. Sizes, fabrication process. It is clear that can modulate at 100Hz but how much power is wasted when driving the emitters? Is there any possibility of future fabrication of this light-controlled modulator in the emitters? That should be discussed in more detail.

We have added a detailed description of the circuit board as a separate section in the Supplementary Information (Supplementary Note 3), along with detailed images of the circuits in Fig. S1. We have also added Fig. S8 in the Supplementary Information which provides a more detailed schematic of the electrical circuit designed to implement the optical control to transducer array, as discussed in the response to Comments 2.3-6 below.

We have further added a detailed power analysis in Supplementary Note 4, accounting for both optical power projected onto the photoresistors as well as the electrical power drawn from the LAPS circuit to perform the operation of phase modulation. We measure an RMS optical power consumption of 23 mW per pixel during maximum illumination and a maximum RMS electrical power consumption of 7 mW per pixel during an active pulse. In comparison, the RMS power drawn by a single pixel of the 11x11 transducer array is between 17-27 mW. We now highlight these results in the main manuscript:

(L243) *The power consumption per transducer element is notably low, requiring a maximum of 30 mW per channel, which is effectively equal to the acoustic power transmitted by each transducer pixel. Of this power, 77% is used for the optical illumination, which we did not optimize (see Supplementary Note 4 for details). Therefore, it is likely that with further design adaptations, such as integrated optical illumination could significantly reduce the parasitic power required by our phase shifting hardware.*

The reviewer raises a very interesting prospect of integrating the modulator directly with the emitters, which could be one pathway to scaling up the number of transmitting elements in the array. While we believe there is value in our current architecture already because it removes the cost and space associated with per-channel driving circuitry, it is indeed possible to envision that an integrated approach could be developed in the future. For instance, a micro LED display could be directly coupled to an integrated version of our analog circuit array with a microlens array for efficient light collection. This circuit could itself be directly connected to a transducer via flip-chip bonding (as in e.g. doi:10.1109/TUFFC.2008.652; doi:10.1109/IUS52206.2021.9593776). Such an architecture would then only require the amplified driving signals and control signals for the display to be transmitted along a cable, allowing for a more compact and potentially flexible holographic projector. We have added these ideas for future integration and scaling down of the circuitry to the discussion section:

(L313) *To effectively drive larger arrays with a compact and flexible footprint, the optical modulator could be integrated, for instance with the help of micro LED arrays in conjunction with a microlens array for efficient light collection. The modulator circuit could then be directly integrated with an emitter transducer via flip-chip bonding [doi:10.1109/TUFFC.2008.652; doi:10.1109/IUS52206.2021.9593776]. Such an integrated architecture could then support further scaling without requiring more complex interconnects and electronics...*

Reviewer #2

This manuscript describes a novel way of optically modulating an ultrasound matrix array. It has the advantage of supplying a single drive waveform which can be optically modulated by a digital projector. It is well written and explanations are clear in general. Some experimental details are missing.

2.1) lines 36-38: The authors may be unaware of matrix arrays using thousands of elements using micro-beamformers and advanced techniques. These methods are described on Philips Ultrasound website and in Chapters 7 and 10 of the book Diagnostic Ultrasound Imaging: Inside Out, (2014)

We thank the reviewer for these comments.

Micro-beamforming in ultrasound is a process where initial beamforming—applying precise time delays and summing signals from small groups of transducer elements—is performed within the probe itself to reduce data bandwidth and improve real-time imaging capabilities. The integration of beamforming circuitry within the probe have been demonstrated in transmit and in receive (doi: 10.1109/UFFC-JS60046.2024.10793797). In transmit, signals from multiple elements are combined with appropriate time delays to focus and steer the ultrasound beam. In receive, beamforming operations are applied on back scattered echoes from a group of elements rather than sending all raw signals to the main system to compute an image. This method reduces the number of data channels required, simplifies cabling, and allows for more compact imaging systems. However, as any standard phased-array, such systems are not geared towards projecting complex field patterns, particularly not with higher powers or longer pulses. Moreover, they cannot provide continuous phase modulation for arbitrary wave driving which is crucially important to generate complex field patterns.

As for the advanced techniques, the reviewer maybe refers to cable count reduction techniques to perform high framerate volumetric imaging such as row-column arrays (doi: 10.1109/ULTSYM.2003.1293560) and sparse arrays (doi: 10.1109/58.8427). Fewer electronic channels are then required to drive all elements. However, this approach loses degrees of freedom to transmit field patterns and thus cannot easily be translated to project complex fields.

We have mentioned the techniques and the corresponding references to these 2 techniques – micro-beamforming and reduction techniques – with the following clarifications added to the introduction of the revised manuscript:

(L54) *Recent technical developments such as micro-beamforming row-column addressing, or sparse arrays, have significantly improved the ability to address more pixels in phased arrays. However, these techniques inherently remove degrees of freedom to transmit field patterns and thus cannot easily be translated to project complex fields for holography.*

2.2) Line 79 reference [24] does not describe dispersion correction but dispersion measurement. An explanation of absorption and dispersion listing original references can be found in Chap. 4 of the previously mentioned book.

Thank you. We have removed this reference and discussion of time reversal/dispersion compensation from the updated manuscript, as our architecture does not currently support frequency-specific phase shifting in broadband pulses (see response to comment 3.2 below).

2.3) Lines 124-131, Equation 1, Fig. 2, line 323 [34] The assumption that the transducer impedance can be represented by a capacitor is not accepted as a valid equivalent circuit model. Ref [34] is only valid for a piezoelectric in air and not acoustically loaded. In general, a piezoelectric transducer radiates into acoustic loads such as those used in both experimental arrays and described in lines 279-282, 333-344.

2.4) Experimentally validated equivalent circuit models such as the Mason or KLM models are explained in Acoustic Waves by Kino or chapter 5 of the previously mentioned book. These models support acoustic loading on both sides, dielectric and absorption losses and harmonics (multiples of the fundamental frequency).

2.5) In general transducer impedances are complex functions of frequency and have real and imaginary parts. Equation 1, however, appears to assume the transducer impedance is a pure capacitor. Whether this is a reasonable approximation in general remains to be proven. Can you show that this approximation holds for the specific transducers in your experiments? Is this assumption reasonable for very small matrix elements?

2.6) The circuit in Fig 2 and Equation 1 and supplement 1 can be adapted to be more realistic by adding a real component R_T .

(Response 2.3-2.6) Given that comments 2.3-2.6 all concern the circuit model for our transducers, we will address the comments together here.

We have updated Fig. 2 to indicate that the impedance of the transducer can in general be complex and also made the corresponding correction in the text, as well as in the caption. We have added three new figures (Fig. S4, Fig. S5, and Fig. S6) depicting a complete transducer equivalent circuit that we discuss in Supplementary Note 2 of the Supplementary Information. We have also included the measurement of the complex impedance corresponding to our transducer elements in the Supplementary Information. These measurements reveal that the transducer response is predominantly capacitive for one-sided loading (Fig. S4 and Fig. S5). Therefore, we have left this approximation in the main text, but added the detailed explanation in the Supplementary Information for better understanding. We thank the reviewer for the suggestion and have added a reference to the book by Szabo in a brief discussion about the transducer impedance in the main text, along with a reference to our added Supplementary Note and Figures.

(L149) *In general, the impedance of a piezoelectric transducer is complex. However, based on measurements on resonance (Supplementary Note 2 and Figs. S4-S5), we approximate the magnitude of complex impedance of each transducer element as a purely capacitive reactance ($|Z_T| \approx |X_T|$).*

2.7) Lines 322-329: Based on the assumption that transducer impedances can be solely represented by capacitors, tests were run on thin film capacitors at different frequencies. These would not constitute a comprehensive study, given that transducers come in many sizes and frequencies and actually have complex impedances. Can you provide more specifics on your experiments like the range of capacitances and frequencies studied? Are these experiments supporting the claim that the circuit supports frequencies from 100kHz to 10 MHz? Furthermore, are these experiments the source of data shown in Fig. 2C? These connections are not clearly stated in the manuscript.

We are sorry that the explanations were not clear regarding the experiments presented in Fig. 2C. The reviewer is absolutely correct in suggesting that we use these measurements to support our claim that the phase-shifting circuit can support different transducers (by varying the capacitive load), and can operate in wide frequencies between 100 kHz – 10 MHz. Since our measurements revealed a purely capacitive transducer response, we tested purely capacitive loads up to 500pF. For Fig 2C, we kept the capacitance constant at 300pF and measured the total electrical phase shift generated by the circuit as a function of driving frequency. We have updated the main text to clarify these experiments and their meaning:

(L183) *The performance of the phase shifter is independent of capacitive loading, making it effective for driving different transducers in different frequency ranges. In order to model the effect of different transducers, we numerically model the circuit response with different ranges of load capacitance up to 500pF, and found that the total phase shift remains constant (Fig. S3 of the Supplementary Information). Then, to verify the ability of the circuit to apply a 2π phase shift for different transducers across a wide range of center frequencies, we measure the phase shifting bandwidth by substituting the transducer element by an equivalent non-polarized thin film capacitor. The LAPS circuit provides a total phase shift of 2π in a wide frequency band ranging from 0.1 MHz to 10 MHz as demonstrated in Fig. 2(C). Thus, the device facilitates controlled realization of sophisticated wavefronts by simple light illumination, with the benefits of analog control of the phase shift and complete phase modulation from $-\pi$ to π .*

2.8) Lines 159-165: The rise and fall times of the proposed method indicate that a 100 Hz frame rate or modulation replenishment can be achieved. The examples in the manuscript are implemented as a quasi-continuous wave methodology. Lines 286-287 confirm this as a tone burst of 25 cycles is used as a drive pulse. It is not clearly explained how broadband width time signals as stated in lines 75-79 can be achieved with these rise and fall times. Please elaborate.

The reviewer is correct that we have primarily focused on quasi-cw transmission because this is closely related to many applications of interest for acoustic holograms. To address the broadband performance of our circuits we have carried out additional experiments transmitting short pulses through our circuit. The results are included in Supplementary Note 7 and Fig. S14. We observe that the circuit provides a nearly constant phase shift across a wide frequency band, as expected, and therefore could evenly delay the phases of pulses with frequency content inside this band. However, as we discuss in our responses to comments 3.1 and 3.2, this does not mean that we can temporally reshape or fully time-shift broadband pulses, because our circuits lack the ability to independently and simultaneously modulate the different frequency components in a broad-band pulse. For further discussion and an overview of the related changes to the text, we refer the reviewer to the discussion to points 3.1 and 3.2 below.

Reviewer #3

This is a really important contribution to the field of acoustic – and I don't say this lightly – particular for HIFU and object manipulation applications. Ever since the development of arrays and their widespread adoption, this issue of scalability has been a major limitation. The issue

is that every element needs a new channel of electronics. Hence, cost scales almost linearly with number of elements. The present paper succeeds in overcoming this limitation and their device uses a single ultrasonic reference input along with optically controllable delays. This means that shaped light (from a light projector) can be used to create user-defined phase patterns. The “pixels” are then light sensitive resistors, which through a simple circuit, produce a phase delay on a piezoelectric element. The response time is fast (c. 5ms), leading to a demonstration that is claimed to have a 100Hz update rate. The results are clearly presented and show that the system operates as intended. I strongly recommend this work is rapidly published. I have a few minor points that need further clarification.

3.1) From Fig 1c it seems that the device is a phase-shifting device. It has this attribute over a wide range of frequencies. This is exactly what is needed for continuous signals or very narrow-band pulses. However, to delay a broad-band pulse requires a phase ramp as the different frequencies need to be delayed by different amounts (of phase) to keep the pulse compact. Hence, whilst the device has broad-band phase-shifting performance, it is not suitable for broad-band pulses. This distinction should be made.

3.2) Related to the above, I am unsure if the device can (as is suggested on P3) be used for time reversal? I think the caveat is needed that the signals being time reversed are narrow-band.

(Response 3.1 & 3.2) Thank you for the comments. The reviewer is correct that our circuit is a phase shifter and therefore can provide a fixed phase shift over a wide range of frequencies (as shown in Fig. 2C). This, however, does not directly address the performance of the circuit for broadband pulses. To assess this, we performed additional experiments transmitting a single-cycle pulse through our array both with and without the phase shifting circuit. We observe that the circuit applies a nearly constant phase shift across the operating band 0.2 MHz – 1.1 MHz. Therefore, we conclude that the OPAT can be used to transmit and phase shift broadband pulses with energy concentrated within this band. Wider-band pulses can be somewhat phase-shifted, although performance outside of this band will lead to distortions of the pulse shape. We have added these results in Fig. S14. We briefly comment on these results in the main text as well in the main text:

(L271) *Such phase shifting performance extends over a wide frequency range as well, allowing for the transmission and phase shifting of relatively short pulses. By transmitting single-cycle pulses through the array, we observe that the circuit applies a nearly constant phase shift across the operating band 0.2 MHz – 1.1 MHz. Therefore, the OPAT can be used to transmit and apply a constant phase shift to broadband pulses with energy concentrated in this band (see Supplementary Note 7 and Fig. S14 for further details).*

As the reviewer noted, however, this does not mean that we can temporally reshape or fully time-shift broadband pulses, because our circuit is not designed to independently and simultaneously modulate the different frequency components in a broad-band pulse. This system is therefore not suited for arbitrary time reversal - we have therefore removed these claims from the revised manuscript.

3.3) P5 – it is said that $65.7\text{mW}/\text{cm}^2$ of light is needed. Can the authors clarify if this is a safe level and perhaps provide some reference that readers can quickly follow, e.g. is this equivalent to a small hand torch?

3.4) Related to the above, if it is too sensitive then it might be hard to use in an environment with natural lighting, can the authors comment?

(Response 3.3 & 3.4) Indeed, ambient light can easily provide enough light to influence the circuits. For reference, direct sunlight irradiates the earth's surface with ca. $100\text{ mW}/\text{cm}^2$ of light (doi: 10.1038/339198a0). While such intensities are safe for skin exposure, staring into such a light source is not recommended. In our experiments, we enclosed the projector and circuits in a dark box and kept the lab lights off to minimize the effects of ambient light. In real applications, we anticipate that the light source and circuit can be integrated into an enclosed box to both minimize the interference from stray light and to protect users from unnecessary direct light exposure.

We have added a brief discussion of these points to the revised manuscript:

(L216) *Relatively low light intensities are needed, as even ambient light can activate photoresistors. The light intensities that are employed in the current system are smaller compared to the intensities of direct sunlight (doi: 10.1038/339198a0). Nevertheless, we expect that a device based on our technique will contain an enclosed illumination setup. While we use a projector, future developments could make use of micro-LED and microlens arrays which would reduce the required light intensities and also permit for more compact designs.*

3.5) In figure 3 (and possibly fig 6) is it possible to quantify the phases obtained? Clearly there are observable phase differences in 3d, but there is also considerable variation. Can this be quantified and its source commented on?

We believe the variation in output phases within the circuit region of uniform illumination is attributable primarily to diffractive effects arising from the distance between the hydrophone and the transducer array

To clarify this point, we performed additional experiments: we projected a square pattern of light onto the circuit and measured the transducer output. The square pattern was made from 5×5 elements which is a sub-array of 11×11 transducer array employed in the experiments. As can be seen in Fig. S13, a geometric pattern is visible associated with interference fringes from the edge of the squares, particularly for a π phase shift. This arises because our hydrophone was placed a finite distance away from the transducer which is approximately 2 mm, leading to interference fringes associated with the projected phase patterns.

We graphically quantify the total phase variation measured in the illuminated regions in Fig. S13, and now comment on this artifact in the revised main text:

(L211) *Minor spatial variations within uniform phase regions can be observed in Figs. 3 (B) and (C), which we attribute to interference and diffraction between transducer output and the hydrophone (see Supplementary Note 6 and Fig. S13 for more details).*

3.6) There are a few very nice schematics of the apparatus, but it would be beneficial to have a photograph so the reader can understand the reality of the set-up.

Thank you very much for the constructive suggestion. We have added overview photographs of the experimental setup in newly added figures Fig. S7, Fig. S8, and Fig. S10 in the Supplementary Information.

We thank the reviewers for supporting the publication of the manuscript. We provide a response to the remarks below.

Reviewer #1 (Remarks to the Author):

The authors have addressed all my questions. I am not fully convinced that the proposed system should go into Nature Comms. In my view, this approach only solves the driving electronics, the wiring and emitters remain with the same issues. I still think existing electronic system can also be scalable and present multiple advantages compared to this system. In any case, it can result of interest to a wide variety of researchers giving that it is easy to replicate and applicable to different fields.

We thank the reviewer for the positive appraisal of our revised manuscript and for the many helpful comments.

Reviewer #2 (Remarks to the Author):

The authors have addressed reviewer comments well.

We thank the reviewer once again for the many comments that have helped improve our manuscript.

Reviewer #3 (Remarks to the Author):

The authors have responded comprehensively to my comments, which were minor issues. Hence I am satisfied with the response and look forward to seeing this important work published. As I mentioned in my original review the importance of this work comes from the breaking of the current link between separate excitation channels and the number of array elements. This link means that as the number of array elements increases, so the cost goes up almost linearly. There are 1000-element systems in existence, but they are prohibitively expensive, and this limits their range of uses. With the concept described in this paper, the number of elements can increase to these levels without the commensurate increase in cost. Hence this opens up the possibility of the use of arrays with many elements in a much wider range of applications. For example, there are many uses of arrays in biomedical applications that are not being explored as the cost of entry is too high. A lower cost array system (using the ideas of this paper) would mean that this technology could be taken up by a much greater range of biomedical researchers.

We thank the reviewer once again for his positive comments on our revised manuscript and for the helpful suggestions to further improve the presentation of our work.